# Analysis of Surface Texture and Roughness in Composites Stiffening Ribs Formed by SPIF Process

**DOI:** 10.3390/ma16072901

**Published:** 2023-04-06

**Authors:** Raheem Al-Sabur, Andrzej Kubit, Hassanein I. Khalaf, Wojciech Jurczak, Andrzej Dzierwa, Marcin Korzeniowski

**Affiliations:** 1Mechanical Department, Engineering College, University of Basrah, Basrah 61004, Iraq; 2Department of Manufacturing and Production Engineering, Faculty of Mechanical Engineering and Aeronautics, Rzeszow University of Technology, Al. Powst. Warszawy 8, 35-959 Rzeszów, Poland; 3Mechanical and Electrical Engineering Department, Polish Naval Academy, 81-103 Gdynia, Poland; 4Department of Metal Forming, Welding and Metrology, Faculty of Mechanical Engineering, Wroclaw University of Science and Technology, Wyb. Wyspianskiego 27, 50-370 Wroclaw, Poland

**Keywords:** roughness, topography, SPIF, LITECOR^®^, single point incremental

## Abstract

Studying roughness parameters and the topography of stiffening ribs in composite sandwich structures is important for understanding these materials’ surface quality and mechanical properties. The roughness parameters describe the micro-geometry of the surface, including the average height deviation, roughness depth, and waviness. The topography of the surface refers to the spatial arrangement and distribution of features such as bumps, ridges, and valleys. The study investigated the roughness parameters under three scenarios based on two SPIF process parameters: tool rotational speed(N) and feed rate (f). The vertical step was held constant at 0.4 mm across all scenarios. In scenario A, the process parameters were set at f = 300 mm/min and n = 300 rpm; in scenario B, f = 1500 mm/min and n = 3000 rpm; and in scenario C, f = 1500 mm/min and n = 300 rpm. The experimental research topography analyses revealed that the surface roughness of the stiffened ribs was highly dependent on the SPIF process parameters. The highest feed rate and tool rotational speed produced the smoothest surface texture with the lowest maximum height (Sz) value. In contrast, the lowest feed rate and tool rotational speed resulted in a rougher surface texture with a higher maximum height (Sz) value. Furthermore, the contour plots generated from the topography analyses provided a good visual representation of the surface texture and roughness, allowing for a more comprehensive analysis of the SPIF process parameters. This study emphasizes optimizing the SPIF process parameters to achieve the desired surface quality and texture of stiffened ribs formed in Litecor^®^ panel sheets.

## 1. Introduction

Metal–polymer–metal (MPM) composites are composite materials of two metal layers with a polymer layer sandwiched in between. This composite material offers several advantages over traditional metal materials, including improved mechanical properties and corrosion resistance [1]. One key advantage of MPM composites is their high strength-to-weight ratio, making them ideal for applications requiring weight and high-performance materials [2,3]. The polymer layer in MPM composites can also help to dampen vibrations and reduce noise, which is particularly important in aerospace applications [4]. MPM composites have become increasingly popular in many industries, including aerospace, automotive, and construction. In aerospace applications, MPM composites are used in various components, such as wing skins, fuselage panels, and control surfaces [5]. Recently, many commercial MPM products have been widely used, such as Alusion™ (Cymat Technologies Ltd., Mississauga, ON, Canada), BONDED METAL™ (Forms+Surfaces, Pittsburgh, PA, USA), ALPOLIC^®^/fr LT (Mitsubishi Chemical Composites America, Inc, Chesapeake, VA, USA), Metalite™ (Coenco, Bangkok, Thailand), KAPPAFLEX™ by Kapp Aluminium GmbH (Bielefeld, Germany), and LITECOR^®^ (ThyssenKrupp, Essen, Germany). LITECOR^®^ is a high-performance composite panel produced by ThyssenKrupp Steel Europe AG. The panel consists of a polyamide (P.A.)/polyethene (P.E.) core sandwiched between two steel layers [6,7]. The steel layers are made of HX220YD, a high-strength steel with excellent strength and stiffness while remaining lightweight. The intermediate layer is a thermoplastic polymer blend that helps to bond the steel layers together and provides additional impact resistance.

Stiffening ribs play a crucial role in enhancing the mechanical properties of composite structures. They increase the material’s overall stiffness, making it more resistant to bending and buckling under load, improving strength and durability [8]. Stiffening ribs provide structural support, enhance mechanical properties, and control a composite structure’s overall shape and contour because the placement and orientation of the ribs can determine the distribution of stresses and strains within the composite material, affecting the structure’s final shape. By strategically placing the ribs, it is possible to control the deformation of the composite structure under load and achieve the desired shape. Furthermore, adding short carbon fibers as a reinforcement material can further increase the composite structure’s strength and stiffness [9]. In addition to the mechanical benefits, stiffening ribs can control a composite structure’s overall shape and contour [10]. By strategically placing stiffening ribs, it is possible to alter the curvature of a composite structure, which can be useful in applications where aerodynamics and hydrodynamics are important factors. For example, in the aerospace industry, stiffening ribs are used to control the shape of aircraft wings, which affects the lift and drag characteristics of the aircraft. Similarly, in the marine industry, stiffening ribs control the shape of boat hulls, affecting their hydrodynamic performance in the water.

Single point incremental forming (SPIF) is a technology for manufacturing stiffening ribs for sandwich structures in composite materials. It is a relatively new forming technique developed in the early 1990s [11]. The method was initially developed for prototyping purposes, but it has since gained popularity in the aerospace, automotive, and biomedical industries for low-volume production of complex-shaped parts [12,13]. Overall, SPIF has emerged as a promising forming technique for the production of complex-shaped parts, with potential advantages in terms of cost, lead time, and design flexibility [14]. SPIF is a sheet metal forming process involving a CNC-controlled single-point tool to gradually shape a flat sheet into a 3D geometry. The process starts with a flat sheet of metal clamped to a forming table. Typically mounted on a CNC-controlled arm, the tool moves in a series of incremental steps to shape the sheet metal into the desired shape [15]. The tooltip contacts the sheet metal and applies pressure, causing the metal to deform and stretch. The tool moves in a series of small incremental steps, each of which causes a localized deformation of the sheet metal. The material is stretched and thinned in the contact areas between the tool and the sheet metal, leading to the desired shape [16]. The process is characterized by the absence of a die or mould, which allows for rapid prototyping and low-volume production runs.

The roughness of stiffening ribs refers to the texture or surface characteristics of the ribs [17]. When a stiffening rib is manufactured, it can have a variety of surface finishes, ranging from very smooth to very rough. A rib’s roughness can affect its mechanical properties, such as strength, stiffness, and fatigue resistance [18].

The concentrated deformation zone distinguishes the SPIF forming process when the tool face comes into contact with the formed sheet. The extent of this zone is determined by the tool end’s size and shape, significantly impacting the sheet metal’s formability and surface characteristics.

The feed rate of the tool, rotational speed, and vertical step size all significantly impact the formability of ISFed sheets, as demonstrated in studies by Buff et al. [19]. The vertical step size can significantly affect both the formability and roughness of the SPIF forming process [20,21]. Generally, smaller step sizes result in more accurate and detailed parts with higher formability. However, smaller step sizes also increase the number of tool passes required to form the part, which can increase cycle time and may result in greater tool wear [22]. Tomasz Trzepiecin’ski et al. [23] studied the effect of SPIS process parameters on the surface roughness of the rib-strengthened aluminium alloy 2024-T3. The results showed that the average roughness parameter (Sa) is proportional to tool rotational speed, while the peak–valley surface roughness (Sz) is inversely proportional to rotational speed. Reducing rotational tool speed and feed rate can improve the inner surface roughness of a machined component. When the tool rotates at a slower speed and moves at a slower feed rate, it allows for greater control and precision in the cutting process, reducing the likelihood of creating surface defects [24]. Surface roughness in SPIFed draw pieces is largely affected by lubrication conditions and tool rotational speed, with tool path, step size, and feed rate playing a minimal role. Therefore, optimizing lubrication conditions and tool rotational speed can lead to reduced surface roughness [25].

Tomasz Trzepieci’nski et al. [26] examined the surface roughness parameters on both sides of pure titanium sheets that SPIS produced. They demonstrated that the feed rate and step size deliver the major effects concerning the kurtosis (Sku) and skewness (Ssk) parameters of the inner surface of the drawing piece. Regarding the drawing piece’s outer surface, the tool rotation direction is nearly associated with Skue, Sz, and Sa.

Studying the effect of tool rotational speed and feed rate on the roughness parameters and topography of stiffening ribs in composite sandwich structures is a novel area of research that can advance our understanding of composite materials manufacturing processes and mechanical behaviour. While there have been numerous studies on the effect of rotational tool speed and feed rate on surface roughness in machining applications, there is limited research on their impact on the surface quality of composite sandwich structures, particularly on the stiffening ribs.

Overall, the novelty of this research lies in its potential to advance the state-of-the-art in the manufacturing and performance of composite sandwich structures through a better understanding of the impact of tool rotational speed and feed rate on the surface quality of stiffening ribs.

## 2. Materials and Methods

In this study, stiffening ribs of LITECOR^®^ composite panels (ThyssenKrupp Steel Europe, Duisburg, Germany) were prepared using the SPIF process by a TM-1P milling machine (Hass Automation, Oxnard, CA, USA), steel pin HS2-9-2, and lubricant SAE 75W-85 (Mannol, Germany). Then, the topographical data processing (complying with ISO 25178 [27]) was performed using Altimap Premium software (Altisurf Company, Rhône-Alpes, France). The Department of Materials Science at Rzeszów University of Technology carried out the experimental work.

### 2.1. Incremental Forming

The composite sandwich panel utilized in this study is LITECOR^®^, as depicted in Figure 1. Table 1 provides an overview of the mechanical properties of LITECOR^®^ [8].

For this study, LITECOR^®^ composite panels were utilized, with a plastic core thickness of 1.3 mm and steel covers with a consistent thickness of 0.3 mm. The LITECOR^®^ composite panels used for embossing had 100 mm × 160 mm dimensions. Figure 2 demonstrates the formed stiffening ribs’ 120 mm length and 20 mm width. The depth of embossing (D) was determined to be 5 mm through experimentation. The research utilized a longitudinal groove SPIF tool to create the necessary forming matrix, with the initial parameters informed by previous studies [28,29].

The forming process utilized a continuous spiral-shaped toolpath with a vertical pitch of 0.4 mm. Three cases were examined, each with different feed rate values (f) and tool rotational speeds (N). The three cases were as follows: case A: f = 300 mm/min, N = 300 rpm; Case B: f = 1500 mm/min, N = 3000 rpm; and case C: f = 1500 mm/min, N = 300 rpm.

### 2.2. Roughness Parameters

The ISO 25178 Surface Texture standards represent a collection of international regulations focused on surface roughness analysis. These standards are unique in that they are the first to address the specification and measurement of 3D surface texture, and they define the necessary parameters and specification operators accordingly [27]. The ISO 25178 Surface Texture Standards allow using two distinct evaluation methods: stylus (contact type) and optical probe (non-contact type) [30]. Using a dual-method approach solves the challenges associated with the profile method. Such challenges include variations in measurement results based on the site of measurement and variations related to the direction of scanning.

The 3D surface roughness measurements were carried out by the measuring machine Talysurf CCI Lite 3D (Taylor Hobson, Leicester, UK). Talysurf CCI Lite 3D is a surface metrology instrument that measures surface roughness, waviness, and form per the ISO 25178 standard [31]. It uses a non-contact optical technique to measure the topography of a surface, providing high-resolution 3D surface data [32]. The Talysurf CCI Lite 3D is commonly used in research and development and in quality control applications in aerospace, automotive, and medical device manufacturing industries. The scanned area was 2.8 mm × 3.0 mm (lens 5×). The measurement process involved scanning 1 million points, with different evaluation area sizes designated by the data processing software (Altimap Premium). Altimap Premium is a 3D surface analysis and metrology software developed by Digital Surf, a leading software provider for surface analysis and metrology [33]. The software is designed to work with various surface measurement instruments, including contact profilometers, confocal microscopes, and scanning electron microscopes (SEM), to analyse and visualize 3D surface topography data [34].

Several roughness parameters can be scanned by Talysurf CCI Lite 3D machine and analysed by Altimap Premium. A roughness characteristic that frequently appears in part drawings is known as the arithmetical mean height (Sa), while the root mean square height (Sq) deals with valleys and peaks of moderately smooth surfaces, both measured by (µm). Skewness (Ssk) and Kurtosis (Sku) are dimensionless parameters that better describe the valleys and peaks of the lubricated surfaces’ tribological properties; negative values represent valleys’ positions and indicate good wear resistance and lubrication. In the evaluated area, the maximum peak and maximum pit heights are expressed by (Sp) and (Sv), respectively, and measured in µm. The corresponding equations for each parameter are given in Table 2 [34].

The Litecor^®^ stiffening rib sheets topography analysis was tested according to ISO 25178 using AltiMap Gold software based on the non-contact optical measurement system of Talysurf CCI Lite 3D (Taylor Hobson, England). The basic surface roughness parameters—i.e., the Sz, Sp, Str, Sv, Sq, Sku, Ssk, and Sa—were selected before the primer application. The research examined the roughness parameters in three cases, determined by two SPIF process parameters: tool rotational speed (n) and feed rate (f). Throughout all cases, the vertical step was kept constant at 0.4 mm. The process parameters were configured as f = 300 mm/min and N = 300 rpm (case A); f = 1500 mm/min and N = 3000 rpm (case B). Finally, case C had the process parameters set to f = 1500 mm/min and N = 300 rpm. The primary surface roughness parameters selected based on five profiles with an area of 2.8 mm × 3.0 mm (lens 5×) are listed in Table 3 for each case. The original sheets’ surface roughness topographies for each case are shown in Figure 3.

## 3. Results and Discussions

In roughness measurement and topography analysis, the maximum height (Sz) is a parameter that indicates the height difference between the highest peak and the lowest valley in a given measurement area. According to ISO 25178 and AltiMap software, as indicated in Figure 4, when the value of Sz is larger, the surface has more significant height variations or roughness features. This fact can be useful in describing rougher or more textured surfaces, such as those found in manufacturing or engineering applications.

However, it is important to note that the interpretation of Sz values can depend on the specific application and context. For example, a larger Sz value may be desirable in applications where surface roughness is desired, such as manufacturing or industrial processes. In other applications, such as optics or precision machining, a smaller Sz value may be more desirable for achieving high precision and accuracy.

In the inner surface roughness measurement analysis of the stiffening ribs of the Litecor^®^ panel, the results show that the maximum height (Sz) varies with changes in the rotational speed (N) and feed rate (f).

As indicated in Table 3, the first case with N = 300 rpm and f = 300 mm/min resulted in an Sz of 32.9 μm. In the second case, with N = 1500 rpm and f = 3000 mm/min, the Sz increased to 45.7 μm. In the third case, with N = 1500 rpm and f = 300 mm/min, the Sz decreased to 18.7 μm.

Increasing the rotational speed can increase the SPIF forming force and temperature, leading to more significant surface roughness and a higher maximum height (Sz), which is evident in the second case, where the maximum height increases with a higher rotational speed. When the rotational speed in SPIF is decreased, the forming force and temperature are also reduced. This reduction leads to decreased surface roughness and a lower maximum height (Sz) of the formed part. Decreased rotational speed reduces friction between the tool and the workpiece, generating less heat during the SPIF process. However, this also means that the forming process may take longer due to the slower speed. Thus, selecting the appropriate rotational speed is crucial to achieving the desired properties and quality of the formed composite part.

Increasing the feed rate can decrease the SPIF forming force and temperature, resulting in lower surface roughness and a lower Sz which was evident in the third case, where the maximum height decreased with a higher feed rate.

The results show that rotational speed and feed rate can affect the surface roughness and the resulting maximum height (Sz) in the SPIF process. Finding the optimal combination of (N) and (f) can help achieve the desired surface roughness and maximum height (Sz) for a specific application.

Contour plots are an essential tool in surface roughness and topography analysis, according to ISO 25178. A contour plot is a graphical representation of the 3D topography of a surface. It shows the variation in height across the surface and provides a visual representation of the surface’s texture, roughness, and shape.

The texture of the contour plot in surface roughness and topography analysis can be affected by several factors, including the (f) and (N) during the SPI process, as indicated in Figure 5.

Lower (f) and (N) produce a rougher surface texture with more peaks and valleys, resulting in a more complex contour plot with higher frequency oscillations (Figure 5a). On the other hand, higher (N) and higher (f) can lead to a more significant material removal rate, resulting in a smoother surface texture and contour plot (Figure 5b). However, a higher (N) and lower (f) can also lead to higher temperatures and thermal deformation, resulting in a rougher surface texture and a more complex contour plot (Figure 5c). 

In conclusion, the (f) and (n) are important factors that can affect the texture of the contour plot in surface roughness and topography analysis during the SPIF process. A thorough understanding of the effect of these factors can lead to better control of the SPIF process and improved surface quality.

Figure 6 shows that an increase in the (N) with low (f) can produce a rougher surface and may also result in the tearing of the Litecor^®^ panel sheet during the stiffening rib forming process in SPIF.

The tool’s rotational speed affects the material removal rate and the amount of heat generated during the forming process. The tool generates more heat at high rotational speeds, which can soften the material and cause tearing or deformation. In addition, low feed rates can lead to prolonged contact time between the tool and the material, which can also increase the heat generated and may lead to material tearing.

Furthermore, a rougher surface texture can be generated at high rotational speeds with low feed rates due to more prominent ridges and valleys forming. The rough surface texture can be undesirable for some applications and may require additional processing or finishing.

Identifying the optimal combination of feed rate and tool rotational speed is crucial to achieving the desired surface roughness and preventing material tearing during the SPIF process to avoid tearing in sheets. 

## 4. Conclusions

This study conducted a topography analysis to investigate the effect of SPIF parameters on the surface roughness of stiffened ribs formed in Litecor^®^ panel sheets. Three cases were considered: case A with process parameters set to f = 300 mm/min and N = 300 rpm; case B with parameters set to f = 1500 mm/min and N = 3000 rpm; and case C with parameters set to f = 1500 mm/min and N = 300 rpm.

Based on the experimental research topography analyses, the following conclusions can be drawn:The surfaace roughness of the stiffened ribs formed in Litecor^®^ panel sheets highly depends on the SPIF process parameters, including (f) and (N).Case B, which had the highest (f) and (N), resulted in the smoothest surface texture with the lowest maximum height (Sz) value.Case A, which had the lowest (f) and (N), resulted in a rougher surface texture with a higher maximum height (Sz) value than case B.Case C, which had a low (N) but a high (f), resulted in a surface texture with a high maximum height (Sz) value and a more complex contour plot than cases A and B.The contour plots generated from the topography analyses provided a good visual representation of the surface texture and roughness of the stiffened ribs formed in Litecor^®^ panel sheets, allowing for a more comprehensive analysis of the SPIF process parameters.

This study highlights the importance of optimizing the SPIF process parameters to achieve the desired surface quality and texture of stiffened ribs formed in Litecor^®^ panel sheets.

## Figures and Tables

**Figure 1 materials-16-02901-f001:**
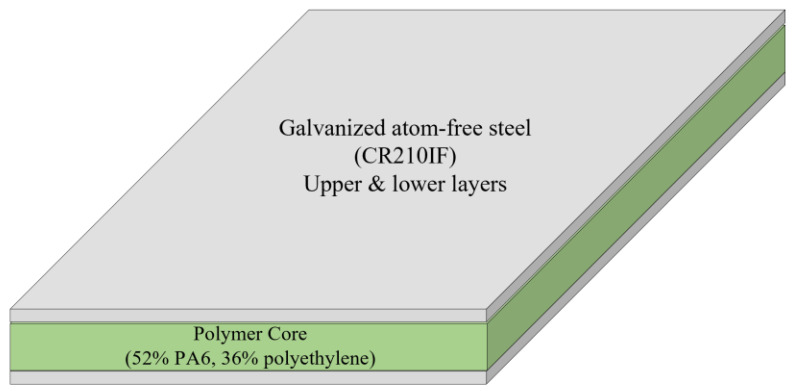
Three layers of LITECOR^®^ panel.

**Figure 2 materials-16-02901-f002:**
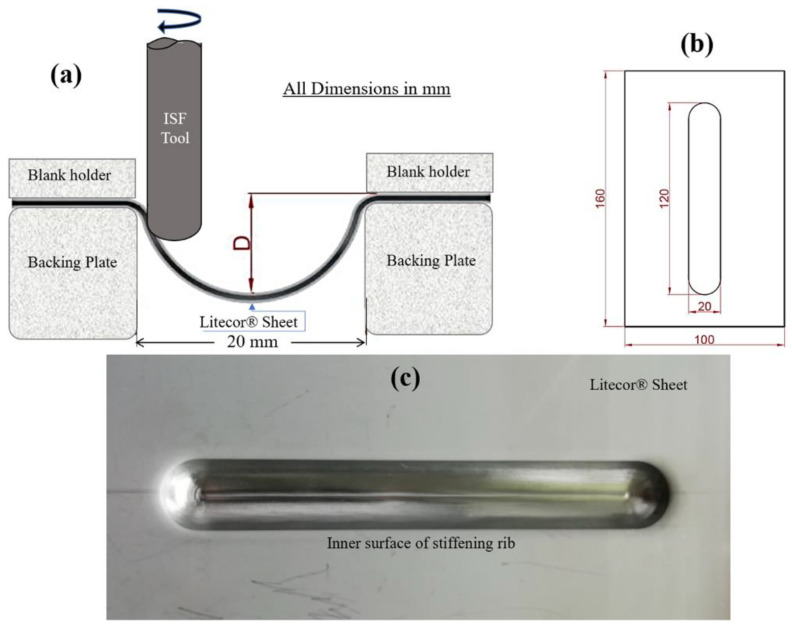
(**a**) SPIF tool scheme, (**b**) sample dimensions, and (**c**) stiffening rib in LITECOR^®^.

**Figure 3 materials-16-02901-f003:**
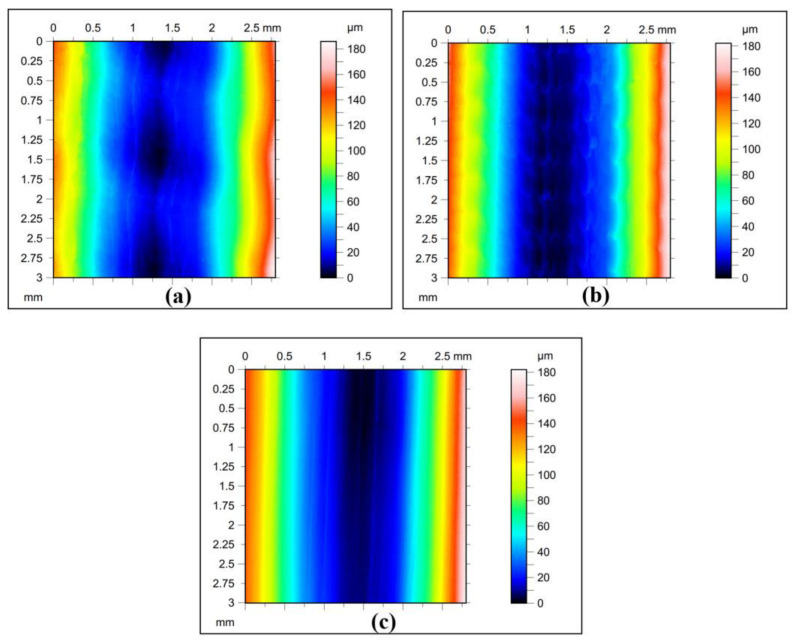
2D profiles of measured sheets at (**a**) case A: f = 300 mm/min, N = 300 rpm (**b**) Case B: f = 1500 mm/min, N = 3000 rpm and (**c**) Case C: f = 1500 mm/min, N = 300 rpm.

**Figure 4 materials-16-02901-f004:**
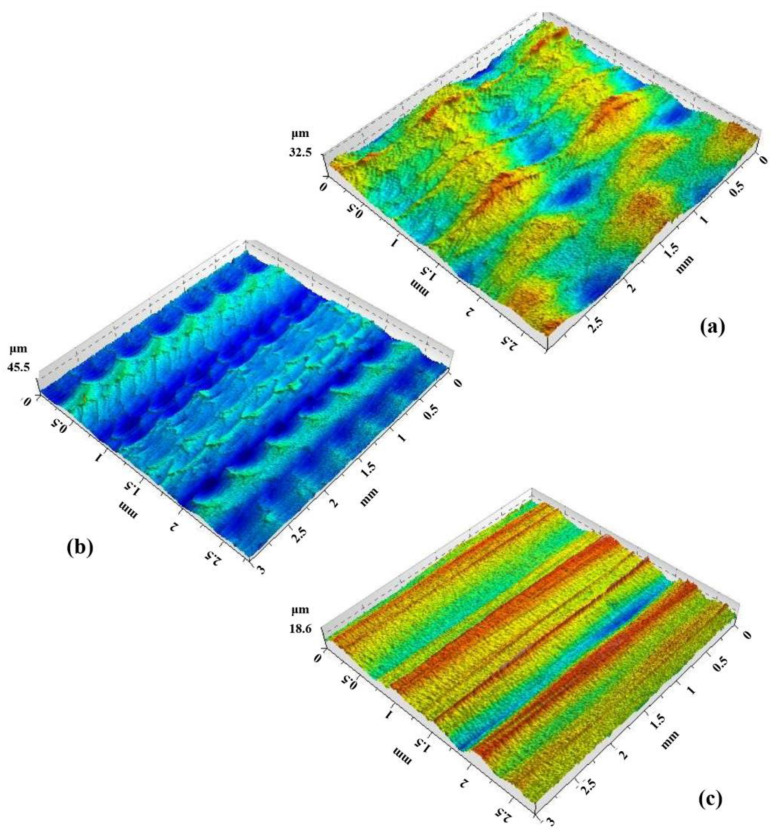
Stiffening rib topographies at (**a**) n = 300 rpm, f = 300 mm/min; (**b**) n = 1500 rpm, f = 3000 mm/min; (**c**) n = 1500 rpm, f = 300 mm/min.

**Figure 5 materials-16-02901-f005:**
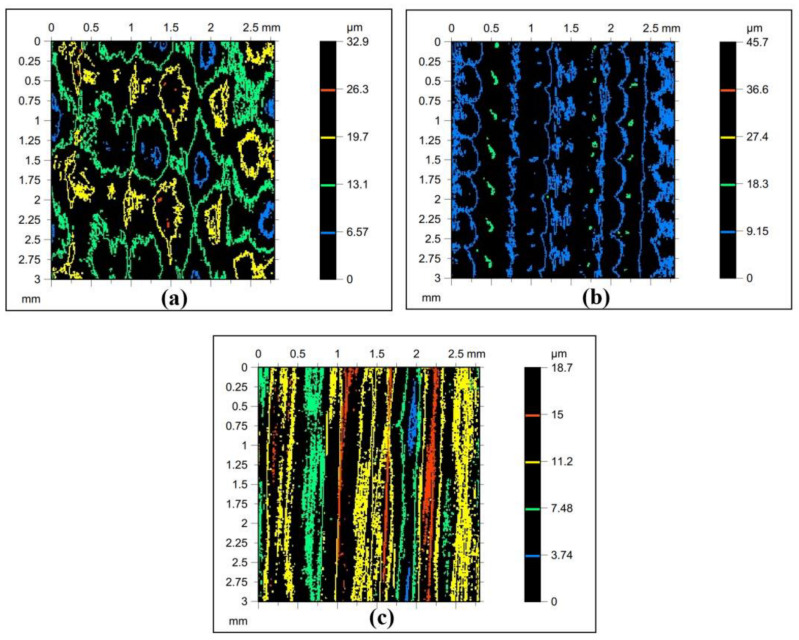
Contour plot of the surface at (**a**) n = 300 rpm, f = 300 mm/min; (**b**) n = 1500 rpm, f = 3000 mm/min; (**c**) n = 1500 rpm, f = 300 mm/min.

**Figure 6 materials-16-02901-f006:**
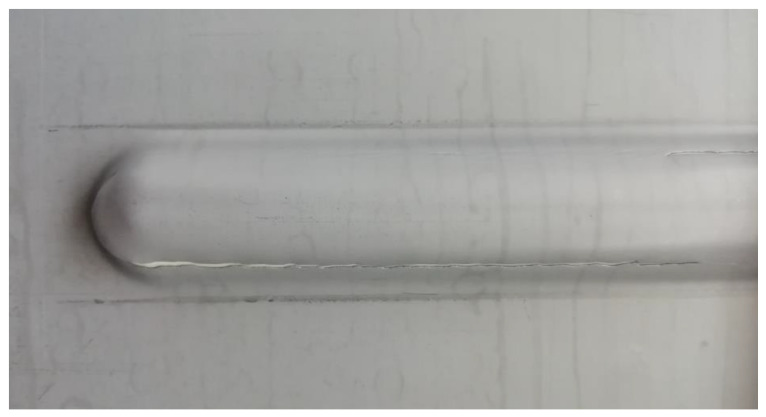
Tearing of the Litecor^®^ ribs surface during case C: f = 1500 mm/min, n = 300 rpm.

**Table 1 materials-16-02901-t001:** Mechanical properties of the LITECOR^®^ panel [8].

Properties	Value
Ultimate Tensile Strength	190–240 MPa
Yield Strength	120–180 MPa
Elongation	28%

**Table 2 materials-16-02901-t002:** Roughness parameters according equation ISO 25178.

Parameter	Equation	Description
Sa	$Sa=\frac{1}{A}\iint_{A}\left|Z\left(x,y\right)\right|dxdy$	Arithmetical mean height
Sq	$Sq=\sqrt{\frac{1}{A}\iint_{A}Z^{2}\left(x,y\right)dxdy}$	Root mean square height
Ssk	$Ssk=\frac{1}{{Sq}^{3}}\left[\frac{1}{A}\iint_{A}Z^{3}\left(x,y\right)dxdy\right]$	Skewness
Sku	$Sku=\frac{1}{{Sq}^{4}}\left[\frac{1}{A}\iint_{A}Z^{4}\left(x,y\right)dxdy\right]$	Kurtosis
Sp	$Sku=\underset{A}{\max}z(x,y)$	Maximum peak height
Sv	$Sv=\underset{A}{\min}z(x,y)$	Maximum pit height
Sz	Sz = Sp + Sv	Maximum height

**Table 3 materials-16-02901-t003:** Litecor^®^ Stiffening ribs sheets roughness parameters.

#	SPIF Parameter		Roughness Parameter							
	F (mm/min)	n (rpm)	Sz (μm)	Sp (μm)	Str	Sv (μm)	Sq	Sku	Ssk	Sa (μm)
Case A	300	300	32.9	18.1	0.159	14.7	4.15	2.61	−0.0439	3.37
Case B	1500	3000	45.7	36	0.101	9.74	3.28	2.62	0.205	2.69
Case C	1500	300	18.7	8.3	0.093	10.4	2.36	2.88	−0.193	1.9

## Data Availability

Not applicable.

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
