# Peer review of "Analysis of Surface Texture and Roughness in Composites Stiffening Ribs Formed by SPIF Process"

_materials, 2023, doi:10.3390/ma16072901_

Round 1
Reviewer 1 Report
In the research article titled “Analysis of Surface Texture and Roughness in Composites Stiffening Ribs Formed by SPIF Process”, presented by Al-Sabur et al. Authors have presented the good analysis, but there are few issues which I think should be addressed. The highlighted issues are as follow;
1. By adding stiffening ribs to a composite structure, the material can resist bending and buckling under load, improving its strength and durability as well as reinforcement material in the form of short carbon fibers. Describe how stiffening ribs can control a composite structure's?
2. Increasing the rotational speed can increase the SPIF forming force and temperature. What effects occurs on the SPIF forming force and temperature if we decrease the rotational speed?
Reviewer 2 Report
The paper investigated the effect of rotational tool speed and feed rate on the roughness parameters and topography of stiffening ribs in composite sandwich structures. I suggest accepting the manuscript after the author has revised it.
1、The sentence expression of the full text needs to be comprehensively improved and polished to avoid some grammatical errors.
2、The text of the picture in the text is too small for readers to read clearly. The format and font size of the figures need to be modified.
3、Roughness parameters and topography of stiffening ribs in composite sandwich structures are important for understanding these materials' mechanical properties. Thus, it is suggested that the author supplement the results of the influence of surface roughness and other parameters on the mechanical properties of composite materials. The content of the current article is insufficient.
Round 2
Reviewer 2 Report
Accept in present form